# SCRIB Is Involved in the Progression of Ovarian Carcinomas in Association with the Factors Linked to Epithelial-to-Mesenchymal Transition and Predicts Shorter Survival of Diagnosed Patients

**DOI:** 10.3390/biom11030405

**Published:** 2021-03-09

**Authors:** Usama Khamis Hussein, Asmaa Gamal Ahmed, Won Ku Choi, Kyoung Min Kim, See-Hyoung Park, Ho Sung Park, Sang Jae Noh, Ho Lee, Myoung Ja Chung, Woo Sung Moon, Myoung Jae Kang, Dong Hyu Cho, Kyu Yun Jang

**Affiliations:** 1Department of Pathology, Jeonbuk National University Medical School, Jeonju 54896, Korea; usamahussein@jbnu.ac.kr (U.K.H.); asmaascience3@gmail.com (A.G.A.); kmkim@jbnu.ac.kr (K.M.K.); hspark@jbnu.ac.kr (H.S.P.); mjchung@jbnu.ac.kr (M.J.C.); mws@jbnu.ac.kr (W.S.M.); mjkang@jbnu.ac.kr (M.J.K.); 2Research Institute of Clinical Medicine of Jeonbuk National University-Biomedical Research Institute of Jeonbuk National University Hospital, Jeonju 54907, Korea; cwkksk@gmail.com; 3Faculty of Science, Beni-Suef University, Beni-Suef 62511, Egypt; 4Faculty of Postgraduate Studies for Advanced Sciences, Beni-Suef University, Beni-Suef 62511, Egypt; 5Department of Obstetrics and Gynecology, Jeonbuk National University Medical School, Jeonju 54896, Korea; 6Department of Bio and Chemical Engineering, Hongik University, Sejong 30016, Korea; imsesame@gmail.com; 7Department of Forensic Medicine, Jeonbuk National University Medical School, Jeonju 54896, Korea; sjnoh@jbnu.ac.kr (S.J.N.); foremed@jbnu.ac.kr (H.L.)

**Keywords:** ovary, cancer, SCRIB, epithelial-to-mesenchymal transition, prognosis

## Abstract

SCRIB is a polarity protein important in maintaining cell junctions. However, recent reports have raised the possibility that SCRIB might have a role in human cancers. Thus, this study evaluated the roles of SCRIB in ovarian cancers. In 102 human ovarian carcinomas, nuclear expression of SCRIB predicted shorter survival of ovarian carcinoma patients, especially in the patients who received post-operative chemotherapy. In SKOV3 and SNU119 ovarian cancer cells, overexpression of SCRIB stimulated the proliferation and invasion of cells. Knockout of SCRIB inhibited in vivo tumor growth of SKOV3 cells and overexpression of SCRIB promoted tumor growth. Overexpression of SCRIB stimulated epithelial-to-mesenchymal transition by increasing the expression of N-cadherin, snail, TGF-β1, and smad2/3, and decreasing the expression of E-cadherin; the converse was observed with inhibition of SCRIB. In conclusion, this study presents the nuclear expression of SCRIB as a prognostic marker of ovarian carcinomas and suggests that SCRIB is involved in the progression of ovarian carcinomas by stimulating proliferation and epithelial-to-mesenchymal transition-related invasiveness.

## 1. Introduction

SCRIB (scribble) is a protein important in maintaining cell polarity and tight junctions of epithelial cells [1]. The role of SCRIB as a component of cell junctions suggests that SCRIB acts as a tumor suppressor because structural and functional alteration of cell polarity induces tumorigenesis [2,3]. Consequently, loss of SCRIB can induce mammary tumorigenesis and promote prostate neoplasia [4,5]. The tumor-suppressive role of SCRIB is also supported by a report that the loss of SCRIB causes loosening of the cell to cell contact and leading epithelial-to-mesenchymal transition (EMT) [1]. However, subsequent reports presented conflicting evidence. Mislocalization of SCRIB from the cytoplasmic membrane to the cytoplasm or nuclei was presented as a tumorigenic phenotype of SCRIB [4,6]. Cytoplasmic enrichment of SCRIB stimulates the EMT pathway and promotes hepatic tumor formation [7]. When considering the significant role of EMT in cancer development and progression [8,9], SCRIB might have a role in cancer progression by regulating EMT [10]. Overexpression of SCRIB stimulated invasiveness of gastric cancer cells by stabilizing β-catenin and stimulating the EMT pathway [10]. Furthermore, elevated expression of SCRIB predicted shorter survival of breast cancer [4] and gastric cancer patients [10]. In addition, SCRIB is involved in the development of B-cell lymphoma stimulated by MYC activation [11]. However, in non-small cell lung cancers, SCRIB expression was associated with a favorable prognosis [12]. Therefore, the effect of SCRIB in tumorigenesis might differ according to its subcellular localization and the molecules acting cooperatively with SCRIB in a specific type of cancer.

Ovarian carcinoma is one of the most common and the most lethal gynecological malignancy [13]. Despite recent advances in the treatment of ovarian carcinomas with specific targeted therapies, especially for angiogenesis and homologous recombination deficiency, the improvement of survival of advanced cancer patients is limited [13]. Tumor stage still remains the most important prognostic indicator [13]. Therefore, more advanced study to find novel therapeutic targets is important to improve the treatment efficacy of advanced ovarian carcinomas. One of the important molecular hallmarks of advanced carcinoma is EMT, and EMT is involved in cancer metastasis and resistance to conventional anti-cancer therapy [9,14,15,16]. Therefore, exploration of EMT-targeted therapy might be helpful in enhancing therapeutic efficacy of advancer carcinomas, including ovarian carcinomas [8,14,15]. Furthermore, recent reports have shown a relationship between SCRIB and EMT in human cancers [7,10]. Thus, the study for SCRIB might be helpful in the understanding of EMT-targeted therapy in ovarian carcinomas. Therefore, we investigated the roles of SCRIB in ovarian carcinomas in conjunction with the EMT phenotype using human ovarian carcinoma tissues and ovarian cancer cells.

## 2. Materials and Methods

### 2.1. Patients and Tissue Samples

This study evaluated 102 ovarian carcinomas diagnosed between November 1996 and August 2008. All cases were reviewed according to the latest WHO classification [17] and the 8th edition of the AJCC (American Joint Committee on Cancer) staging system [18]. The data for clinicopathological factors were obtained by review of medical records and histologic slides. The factors evaluated in this study were the age of patients, preoperative serum level of CA125, tumor size, presence of ascites at diagnosis, tumor stage, lymph node metastasis, bilaterality of tumor, histologic grade, histologic type, and platinum resistance. The cases included in this study (according to the histologic type of ovarian carcinoma) were 73 serous carcinomas (consisting of 11 low-grade serous carcinomas and 62 high-grade serous carcinomas), 20 mucinous carcinomas, 5 endometrioid carcinomas, 3 clear cell carcinomas, and 1 malignant Brenner tumor. Post-operatively, 80 patients received chemotherapy and were evaluated for platinum resistance [19], and 18 patients were platinum-resistant. This study was approved by the institutional review board of Jeonbuk National University Hospital (IRB number, CUH 2019-09-034).

### 2.2. Immunohistochemical Staining in Human Ovarian Carcinoma Tissue

The expression of SCRIB in human ovarian carcinoma tissue was evaluated via immunohistochemical staining of tissue microarray sections. The tissue microarray cores were 5 mm in diameter, and one core per case was arrayed from areas of the highest histologic grade without degenerative or necrotic changes. An antigen retrieval procedure was performed with a microwave oven. The tissue sections were boiled in pH 6.0 antigen retrieval solution (DAKO, Glostrup, Denmark) for 20 min. The primary antibody for SCRIB (1:50, Santa Cruz Biotechnology, Santa Cruz, CA, USA) was used for staining. Immunostained slides were scored with consensus by two pathologists (K.M.K. and K.Y.J.) without clinical information. Based on the subcellular expression patterns of SCRIB in nuclei and cytoplasm in human cancer tissue, immunohistochemical expression of SCRIB was separately analyzed according to its cytoplasmic and nuclear expression patterns. The scoring of immunostained slides was conducted according to the Allred scoring system [20]. The Allred scoring system utilizes a staining intensity score (0; no staining, 1; weak, 2; intermediate, 3; strong) and a staining area score (0; no staining, 1; 1%, 2; 2–10%, 3: 11–33%, 4; 34–66%, 5; 67–100%) [20,21,22]. The final scores were obtained by adding the staining intensity score and staining area score.

### 2.3. Cell Lines and Transfection

SKOV3 (ATCC, Manassas, VA, USA) and SNU119 (KCLB, Seoul, Korea) ovarian cancer cell lines were used in this study. The SKOV3 cell line was grown in DMEM/F-12 (Nutrient Mixture F-12 (Ham) (1:1) powder with L-glutamine, pyridoxine hydrochloride and without HEPES buffer) (Catalog #; 12500-062, GIBCO Invitrogen Corporation, Grand Island, NY, USA), and the SNU119 cell line was grown in RPMI 1640 culture medium (with L-glutamine, 25 mM HEPES buffer and without sodium bicarbonate) (REF. 23400-021, GIBCO Invitrogen Corporation, Grand Island, NY, USA) at 37 °C under humidified conditions with 5% CO_2_. The culture media was supplemented with 10% Fetal Bovine Serum (Gibco BRL, Gaithersburg, MD, USA) and 1% penicillin/streptomycin (Gibco BRL, Gaithersburg, MD, USA). The knockout of *SCRIB* was induced by an hSCRIB/CAS9 KO plasmid (Catalog #; sc-400384, Santa Cruz Biotechnology, Santa Cruz, CA, USA). Overexpression of SCRIB was induced using an overexpression vector for SCRIB (Catalog #; EX-Z2845-M03, accession #; NM_002467, GeneCopoeia, Rockville, MD, USA). The jetPRIME transfection reagent (Polyplus Transfection, Illkirch, France) was used for transfection.

### 2.4. Proliferation Assays

A Cell Counting Kit-8 (CCK8, Dojindo, Kumamoto, Japan) assay and a colony-forming assay were used to evaluate the proliferation of cells. The CCK8 assay was performed by growing 3 × 10^3^ SKOV3 and 3 × 10^3^ SNU119 cells in 96-well plates for 24, 48, and 72 h. At the indicated time, CCK8 was added for two hours, and the absorbance was measured at 560 nm in a microtiter plate reader (Bio-Rad, Richmond, CA, USA). The colony-forming assay was performed by culturing 1 × 10^3^ cells per well in 6-well culture plates. The cultures were grown in triplicate and allowed to grow and form colonies for two weeks. The colonies in the culture plates were fixed with methanol and stained with 0.01% crystal violet. Quantification of the number of colonies was performed using Clono-Counter software (software was downloaded from the supplementary electronic material) [23].

### 2.5. In Vitro Wound Healing and Trans-Chamber Migration and Invasion Assays

The migration and invasion activity of ovarian cancer cells were evaluated with a wound-healing assay and trans-chamber migration and invasion assays. The wound-healing assay was performed by making linear scratches with the tip of a 200 μL pipette in 60 mm culture plates that had reached 100% confluency with cells. The microscopic images for wound healing assay were taken immediately following the application of the linear scratches, and again 24 h after making the scratches. For the migration assay, SKOV3 (5 × 10^4^) and SNU119 (1 × 10^5^) cells were grown for 48 h in the 24-transwell migration chamber (Corning Life Sciences, Acton, MA, USA). Invasion assay was performed by growing SKOV3 (1 × 10^5^) and SNU119 (2 × 10^5^) cells for 48 h in the bioCoat Matrigel Invasion Chamber (Corning 24-Well Plate 8.0 Micron, REF. 354480, ready-to-use, BD Biosciences, San Jose, CA, USA). The migrated and invaded cells on the underside of the insert were counted in five microscopic fields (magnification ×100) per well after being stained with DIFF-Quik staining solutions (Sysmex, Kobe, Japan).

### 2.6. Western Blotting Assay

The cultured cells were washed twice with phosphate buffered saline and lysed using ice-cold PRO-PREP Protein Extraction Solution (iNtRON Biotechnology, Seongnam, Korea) supplemented with 1× phosphatase inhibitor cocktail 2,3 (Sigma, St. Louis, MO, USA). The normalized protein was loaded in a 4× SDS-PAGE, electrophoresed on a SDS-polyacrylamide gel, and electrotransferred to a polyvinylidene difluoride membrane. Anti-SCRIB (Cat. No. #; 4475, Cell Signaling Technology, Beverly, MA, USA), anti-β-catenin (Catalog #; 610154, Millipore, Darmstadt, Germany), anti-active β-catenin (Catalog #; 05-665, Millipore, Darmstadt, Germany), anti-cyclin D1 (Catalog #; 2922, Cell Signaling Technology, Beverly, MA, USA), anti-E-cadherin (Catalog #; 610182, BD Biosciences, Becton, Dickinson), anti-N-cadherin (Catalog #; 13116, Cell Signaling Technology, Beverly, MA, USA), anti-snail (Catalog #; ab180714, Abcam, Cambridge, UK), anti-TGF-β1 (Catalog #; 3709, Cell Signaling Technology, Beverly, MA, USA), anti-smad2/3 (Catalog #; 3102, Cell Signaling Technology, Beverly, MA, USA), and anti-actin (Catalog #; sc-376421, Santa Cruz Biotechnology Inc., Santa Cruz, CA, USA) antibodies were used as the primary antibodies. The membranes were developed with an ECL detection system (Amersham Biosciences, Buckinghamshire, UK). The images were obtained by using a luminescent image analyzer (LAS-3000, Fuji Film, Tokyo, Japan) and quantified using ImageJ software (ImageJ, version 1.38e, NIH, Bethesda, MD, USA).

### 2.7. Quantitative Reverse-Transcription Polymerase Chain Reaction (qRT-PCR)

Total RNA was obtained using an RNeasy Mini Kit (Qiagen Sciences, Valencia, CA, USA). After normalization of RNA, 1.5 μg RNA was reverse transcribed to cDNA using an RT-qPCR kit (Takara Biotechnology Co., Ltd., Dalian, China). A quantitative reverse-transcription polymerase chain reaction was performed with the Applied Biosystems Prism 7900HT sequence Detection System and SYBR Green PCR Master Mix (Applied Biosystems, Foster City, CA, USA). All experiments were performed in triplicate, and the values were normalized to the expression of the *GAPDH* reference gene. To calculate qRT-PCR results, the *delta-delta-Ct* method was used. In this method, we calculate the differences between gene being Tested Experimental (TE) and Housekeeping gene Experimental (HE) [TE-HE], and the differences between gene keeping Tested Control (TC) and Housekeeping gene Control (HC) [TC-HC]. These are the *delta Ct* values for the Experimental (*delta*-CTE or ΔCTE) and Control (*delta*-CTC or ΔCTC) conditions. Finally, we calculate the difference between ΔCTE and ΔCTC (ΔCTE−ΔCTC) to get the *delta-delta-Ct* or what we knew *double delta-Ct* (ΔΔCt) values. All experiments were performed in triplicate. The primer sequences used in the qRT-PCR are listed in Table 1.

### 2.8. In Vivo Tumorigenic Assay

Five-week-old female FoxnN.Cg/c nude mice (Orient Bio, Seongnam, Korea) were used in vivo tumorigenic assay with the approval of the institutional animal care and use committee of Jeonbuk National University (approval number: CBNU 2019-067). The mice were randomly assigned to three groups, with four mice in each group. According to the experimental groups, SKOV3 cells were transfected with empty vector, SCRIB-overexpressing vector, or vector for knock-down of SCRIB. Next, 2 × 10^6^ SKOV3 cells were transfected with the indicated vectors via subcutaneous injection into the flank after mixing with Corning matrigel Basement Membrane Matrix (1:1) (Catalog #; 356234, Bedford, MA, USA). Tumor size was measured every week and calculated by the following equation: tumor volume = length × width × height × 0.52 [10,22,24]. The mice were euthanized according to humane end points at six weeks after tumor cell inoculation. The mice were euthanized after anesthetizing with sodium pentobarbital. The size and weight of the tumors were measured. Hematoxylin and eosin staining was performed on the resected tumor tissue, lung, liver, and kidney.

### 2.9. Statistical Analysis

The immunohistochemical positivity of the nuclear expression of SCRIB (nSCRIB) and cytoplasmic expression of SCRIB (cSCRIB) was determined by receiver operating characteristic (ROC) curve analysis [25]. The cut-off points in ROC curve analysis for nSCRIB and cSCRIB were determined to predict the cancer-related death of patients [10,25,26]. The prognosis of ovarian carcinoma patients was evaluated for overall survival (OS) and relapse-free survival (RFS) through June 2015. An event in OS analysis was the death of a patient from ovarian cancer. An event in RFS analysis was the relapse of cancer and death of a patient from ovarian cancer. Univariate and multivariate Cox proportional hazards regression analysis were used to present a hazard ratio (HR) and 95% confidence interval (95% CI). Survival curves were derived from Kaplan-Meier survival analysis. The relationship and difference between factors were analyzed with the Pearson’s chi-square test and the Student’s *t*-test, and the values are presented as the mean ± standard deviation. All experiments were done in triplicate and performed three times, with representative data presented. SPSS software (IBM, version 20.0, Armonk, NY, USA) was used in statistical analysis, and a *p*-value less than 0.05 was considered statistically significant.

## 3. Results

### 3.1. The Expression of SCRIB Is Associated with the Progression of Ovarian Carcinomas

Immunohistochemically, SCRIB was expressed in both the cytoplasm and nuclei of ovarian carcinoma cells (Figure 1A). As shown in Figure 1A, there were cases that did not express SCRIB, and cases that expressed SCRIB predominantly in the cytoplasm, predominantly in the nuclei, or both the cytoplasmic and the nuclei. Therefore, the expression of SCRIB was separately analyzed according to its cytoplasmic or nuclear expression. The cut-off points of immunohistochemical staining scores for the nuclear expression of SCRIB (nSCRIB) and cytoplasmic expression of SCRIB (cSCRIB) were determined with ROC analysis to predict the death of cancer patients (Figure 1B). The cut-off points were six in both nSCRIB and cSCRIB (Figure 1B). The cases with a score equal or greater than six for both nSCRIB and cSCRIB were considered positive. With these cut-off points, the factors significantly associated with both nSCRIB and cSCRIB were CA125 level, cancer stage, ascites, bilaterality of cancer, histologic grade, histologic type, and platinum resistance (Table 2).

### 3.2. The Expression of SCRIB Is Associated with Shorter Survival of Ovarian Carcinoma Patients in Univariate Analysis

In univariate survival analysis, age, cancer stage, cancer size, presence of ascites, bilaterality of cancer, CA125 level, histologic grade, nSCRIB, and cSCRIB were significantly associated with OS or RFS (Table 3) (Figure 2).

In addition, we performed survival analysis according to histologic subtype of ovarian carcinomas. In high-grade serous carcinoma (HGSC), age, cancer stage, and nSCRIB were significantly associated with OS or RFS in univariate analysis (Table 3) (Figure 3A). In addition, despite the low number of cases, nSCRIB and cSCRIB were significantly associated with survival of mucinous carcinoma patients (Figure 3B).

### 3.3. Nuclear Expression of SCRIB Predicts Shorter Survival of Ovarian Carcinoma Patients in Multivariate Analysis

The factors significantly associated with OS or RFS in univariate analysis except for CA125 level were included in multivariate analysis. In overall 104 ovarian carcinomas, age, cancer stage, histologic grade, and nSCRIB were associated with OS or RFS (Table 4). The nSCRIB positivity predicted a 2.252-fold (95% CI; 1.205–4.207, *p =* 0.011) greater risk of death of patients. In HGSC, age and nSCRIB were independent indicators of OS, and tumor stage was an independent indicator of RFS (Table 4). nSCRIB-positive patients had a 2.236-fold (95% CI; 1.097–4.558, *p =* 0.027) greater risk of death compared with nSCRIB-negative patients with HGSC (Table 4).

### 3.4. The Expression of SCRIB Predicts Shorter Survival of Ovarian Carcinoma Patients Who Received Adjuvant Chemotherapy

In our results, nSCRIB and cSCRIB were significantly associated with platinum-resistance and shorter survival of patients. These results suggest that SCRIB might be related to chemoresistance. Therefore, we further analyzed for survival in the sub-population of ovarian carcinoma patients who received adjuvant chemotherapy. In eighty patients who received adjuvant chemotherapy, both nSCRIB and cSCRIB were significantly associated with OS and RFS in univariate analysis (Table 5) (Figure 4). In multivariate analysis, age, cancer stage, bilaterality, and nSCRIB expression were independent indicators of OS (Table 5). Higher tumor stage and nSCRIB positivity predicted shorter RFS (Table 5). Positivity of nSCRIB predicted a 2.363-fold (95% CI; 1.149–4.859, *p =* 0.019) greater risk of death and a 2.530-fold (95% CI; 1.313–4.875, *p =* 0.006) greater risk of relapse or death of patients (Table 5).

### 3.5. SCRIB Is Associated with the Proliferation and Invasion of Ovarian Cancer Cells

In ovarian cancer cells, we evaluated the effect of SCRIB expression on the proliferation and invasiveness of cells. The proliferation of SKOV3 and SNU119 ovarian cancer cells were significantly inhibited by knockout of *SCRIB* and significantly stimulated by overexpression of *SCRIB* as evidenced by CCK8 proliferation assay (Figure 5A) and colony forming assay (Figure 5B). In addition, the migration and invasion ability of SKOV3 and SNU119 ovarian cancer cells were significantly inhibited by knockout of *SCRIB* and were significantly stimulated by overexpression of *SCRIB* as evidence by wound-healing, migration, and invasion assay (Figure 6). Furthermore, overexpression of SCRIB significantly increased in vivo growth of SKOV3 cells, and knockout of *SCRIB* significantly decreased in vivo tumor growth compared with controls (Figure 7). However, histologically, there were no identifiable metastatic lesions in the lung, liver, or kidney of the mice in the three experimental groups. In addition, there were no significant differences in the shape of implanted cancer cells between experimental groups (Figure 7C).

### 3.6. SCRIB Is Associated with a Change in Expression of Factors Linked to the EMT of Cancer Cells

In malignant epithelial tumors, the molecules associated with EMT are closely associated with cancer progression and resistance to therapy. Therefore, we evaluated the expression of proteins and mRNA transcripts associated with EMT. The protein and mRNA expression of cyclin D1, N-cadherin, snail, TGF-β1, and smad2/3 decreased, and the expression of E-cadherin increased with knockout of *SCRIB* in both SKOV3 and SNU119 cells (Figure 8). Overexpression of *SCRIB* increased the protein and mRNA expression of cyclin D1, N-cadherin, snail, TGF-β1, and smad2/3 and decreased expression of E-cadherin (Figure 8). However, the expression of β-catenin mRNA was unchanged with knockout or overexpression of *SCRIB* despite the change in β-catenin protein expression (Figure 8).

## 4. Discussion

In this study, the expression of SCRIB was associated with advanced clinicopathological factors of ovarian carcinomas, such as elevation of CA125 level, high cancer stage, and higher histologic grade. In addition, both nSCRIB and cSCRIB positivity were independent indicators of shorter OS and RFS. Furthermore, nSCRIB positivity predicted shorter OS of HGSC, the most common histologic type of ovarian cancer with a relatively unfavorable prognosis. These findings suggest that SCRIB is involved in the progression of ovarian carcinomas and might be used as a prognostic marker of ovarian carcinoma patients. Consistently, nSCRIB was an independent predictor of survival of gastric carcinoma patients [10]. In addition, higher expression of SCRIB mRNA was associated with poor prognosis of breast cancers [4] and shorter survival of hepatocellular carcinoma patients [7]. However, controversially, higher immunohistochemical expression of SCRIB was associated with favorable prognosis of lung cancer patients [12]. Therefore, because of controversial reports on the prognostic impact of SCRIB expression in human cancers, further study is needed in various types of human cancers.

Regarding the subcellular localization of SCRIB, its expression was expected in the basolateral side of the cytoplasmic membrane because SCRIB is involved in maintaining tight junctions [1,2,3]. However, the expression of SCRIB was observed in the cytoplasm and/or nuclei of human cancer cells [4,10,27]. The subcellular localization of SCRIB was different between non-neoplastic cells and cancer cells [7,27]. In hepatocytes of healthy livers, SCRIB is localized in the cytoplasmic membrane [7]. However, cytoplasmic localization of SCRIB was detected in 69% (22/32 cases) of hepatocellular carcinomas [7]. Moreover, enrichment of cytoplasmic SCRIB was associated with the overall increase of intracellular SCRIB by inducing overexpression of SCRIB [7]. In a study with human and mouse liver cancer models, cytoplasmic and nuclear expression of SCRIB was more prevalent in liver cancer tissue, and non-neoplastic liver tissue showed SCRIB expressed solely on the cytoplasmic membrane [27]. The cytoplasmic and nuclear expression of SCRIB in liver cancer tissue was supported by Western blots performed with subcellular fractionated protein lysates [27]. Therefore, higher expression of SCRIB supports its translocation to the cytoplasm and nuclei and might be involved in the tumorigenesis of epithelial cells [7]. Furthermore, in our results, nSCRIB is an independent indicator of shorter survival of ovarian carcinoma patients. These results suggest that nuclear SCRIB is important in cancer progression. Thus, the prognostic significance of nuclear localization of SCRIB might be related to the association between SCRIB and nuclear proteins involved in cancer progression, such as snail and β-catenin [8,10]. In gastric carcinomas, SCRIB stabilizes β-catenin by binding it and consequently activating TCF/LEF transcription [10]. In our results, the expression of proteins of β-catenin were decreased with knockout of *SCRIB* and increased with overexpression of SCRIB. In addition, as a consequence of the SCRIB-mediated increase of β-catenin protein, the expression of cyclin D1 and snail, the down-stream molecules of Wnt/β-catenin signaling, were increased with overexpression of SCRIB. Supportively, mislocalization of SCRIB from the cytoplasmic membrane to cytoplasm promoted mammary tumorigenesis by activating the Akt/mTOR pathway [4]. Together, these results suggest that subcellular localization of SCRIB, especially in nuclei and cytoplasm, might be important in the tumorigenic role of SCRIB, which warrants additional study.

Loss of cell polarity protein might serve to induce tumors by promoting the uncontrolled proliferation of cells [28,29]. Therefore, as a component of polarity protein, SCRIB has been suggested as a tumor suppressor, and loss of SCRIB induced tumorigenesis in MYC-induced transformed epithelial cells [6]. However, in addition to the loss of SCRIB, mislocalization of SCRIB also induced mammary tumorigenesis [6]. Furthermore, cytoplasmic and nuclear localization of SCRIB, induced by overexpression of SCRIB, stimulated the proliferation of liver cancer cells [27]. Consistently, in our results, SCRIB positively regulated the proliferation of ovarian cells. In line with these results in ovarian cancers, SCRIB stimulated in vitro proliferation and in vivo growth of gastric cancer cells by activating the Wnt/β-catenin pathway [10]. Based on the association between SCRIB and the Wnt/β-catenin pathway, recent reports have shown the molecular relationship between MYC, FAM83H, SCRIB, and the Wnt/β-catenin pathway in human cancers [10,11,30]. MYC transcriptionally controls FAM83H expression, and the expression of SCRIB was regulated by FAM83H [10,30]. Thereafter, FAM83H and SCRIB form a complex to stabilize β-catenin [10]. These molecular relationships might be supported by the report that SCRIB is involved in the initiation of MYC-driven lymphoma [11]. Knock-out of SCRIB delayed expansion of B cells in the development of MYC-mediated lymphoma development [11]. Therefore, these results suggest there is a close relationship between MYC, FAM83H, SCRIB, and the Wnt/β-catenin pathway in cancer progression. However, there are conflicting reports that SCRIB acts as a tumor suppressor. Conditional knock-out of SCRIB in the epidermis enhanced epidermal carcinogenesis [31] and SCRIB repressed Wnt signaling in HEK293T cells [32]. Therefore, further study is needed to clarify the role of SCRIB in human cancer.

In our results, SCRIB stimulated invasiveness and the factors linked to EMT of ovarian cancer cells. EMT is an important process in cancer progression and is involved in cancer invasiveness, and the characteristic phenotype of EMT is a loss of E-cadherin in epithelial cells and activation of the snail, TGF-β1, and N-cadherin [9,14]. In our results, SCRIB induced an EMT phenotype; decrease of E-cadherin and increase of N-cadherin, snail, TGF-β1, and smad2/3 with overexpression of SCRIB. Interestingly, the expression of the protein of smad2/3 increased with overexpression of SCRIB. When considering the role of TGF-β1 in EMT as an activator of smad2/3 by inducing phosphorylation of smad2/3 [33], the increase in total protein amount of smad2/3 in SCRIB-overexpressing cells is questionable. However, in addition to the increase in smad2/3 protein, mRNA of smad2/3 also increased with overexpression of SCRIB. Therefore, higher expression of smad2/3 protein in SCRIB-overexpressing cells might be related to increased transcription of smad2/3. Consistently, when we searched the GEPIA public database (http://gepia.cancer-pku.cn. accessed 21 February 2021) [34], there was a significant correlation between the expression of TGF-β1 mRNA and smad2 mRNA (Pearson’s *R* = 0.33, *p* < 0.001) and smad3 mRNA (Pearson’s *R* = 0.31, *p* < 0.001) in ovarian cancers. Furthermore, mRNA expression of SCRIB was also significantly correlated with smad2 mRNA (Pearson’s *R* = 0.12, *p* = 0.013) and smad3 mRNA (Pearson’s *R* = 0.17, *p* < 0.001) in ovarian cancers. In addition, SCRIB induced expression of factors linked to EMT phenotype and increased invasion of gastric cancer cells [10] and cytoplasmic SCRIB increased the invasion activity of liver cancer cells [7]. Therefore, these results suggest that SCRIB is involved in the progression of ovarian cancer by inducing the expression of factors linked to EMT. However, SCRIB is a component of the Scribble polarity complex and important in maintaining the apicobasal cell polarity of epithelial cells. Therefore, based on the role of SCRIB as a component of tight junctions, the maintenance of SCRIB is expected as a tumor suppressor or suppressor of EMT [35,36]. In corneal epithelium, conditional deletion of SCRIB induced EMT phenotype; elongation of the shape of epithelial cells and loss of E-cadherin and upregulated expression of snail TGF-β, and smad3/4 [35]. Knock-down of SCRIB in cancer-associated fibroblasts increased invasiveness of both tumor-associated fibroblasts and co-cultured Lewis lung cancer cells [36]. However, in contrast, recent reports have shown that the role of SCRIB in human cancer is closely related to its subcellular localization instead of its expression level. In a study using tissues from normal endometrium, endometriosis, and endometrial adenocarcinoma, membranous expression of SCRIB in normal endometrium changed to be localized in the cytoplasm and nuclei in endometriosis and endometrial carcinomas [37]. In this respect, the loss of SCRIB from the cytoplasmic membrane and its aberrant translocation to the cytoplasm and/or nucleus might serve in the development and progression of cancers. When SCRIB was overexpressed, SCRIB moved to the cytoplasm, and that was associated with cancer development. In addition, transfection of mutant SCRIB caused a cytoplasmic enriched induced EMT phenotype of hepatocellular carcinoma cells and increased invasiveness of cancer cells [7]. Therefore, SCRIB mediated EMT might be related to the subcellular expression pattern of SCRIB, and further study is needed to focus on this point.

One of the interesting findings of our study is that SCRIB might be involved in chemoresistance of ovarian cancer cells. In the comparison of clinicopathologic factors and SCRIB expression in ovarian carcinomas, nSCRIB and cSCRIB positivity were significantly associated with platinum-resistance. Furthermore, in subpopulation survival analysis of patients who received adjuvant chemotherapy, nSCRIB positivity was an independent indicator of shorter survival. In addition, SCRIB expression was closely associated with the expression of EMT-related molecules. The Wnt, TGF-β, Notch, and mitogenic growth factors related pathways are known as the signaling pathways that activate EMT [8,9,14]. Furthermore, EMT mediates chemoresistance by inducing stem-cell-like properties in cancer cells, inducing resistance to drug-induced apoptosis, and modulating tumor microenvironment on cancer-associated fibroblasts and immune cells [14]. In addition, the EMT phenotype was associated with chemoresistance and progression of ovarian cancer cells [15,38]. Therefore, when considering the effect of EMT on cancer drug resistance [8,14,16], SCRIB might be involved in chemoresistance during the treatment of ovarian carcinomas by activating EMT pathways. However, there are controversial reports on the role of SCRIB on EMT and chemoresistance. In corneal epithelium, the EMT phenotype occurred in SCRIB deficient cells [35]. In non-small cell lung cancer cells, SCRIB supported the anti-cancer effects of cisplatin by increasing cisplatin-mediated generation of reactive oxygen species and knock-down of SCRIB induced cisplatin resistance [12]. Therefore, the role of SCRIB in tumorigenesis and cancer progression might be different according to the type of tumor and stage of progression of cancer, and further study is needed.

## 5. Conclusions

In this study, we demonstrated that SCRIB stimulates proliferation and invasiveness of ovarian cancer cells in conjunction with activation of the factors linked to the EMT pathway. Moreover, we presented SCRIB expression as a potential prognostic indicator of ovarian carcinoma patients, especially in patients who received chemotherapy. Therefore, our results suggest that SCRIB is involved in cancer progression and chemoresistance by activating some EMT characteristics. Furthermore, our results suggest that the SCRIB-EMT pathway might be a new therapeutic target for the ovarian carcinomas which highly express SCRIB and are refractory to the conventional anti-cancer therapies.

## Figures and Tables

**Figure 1 biomolecules-11-00405-f001:**
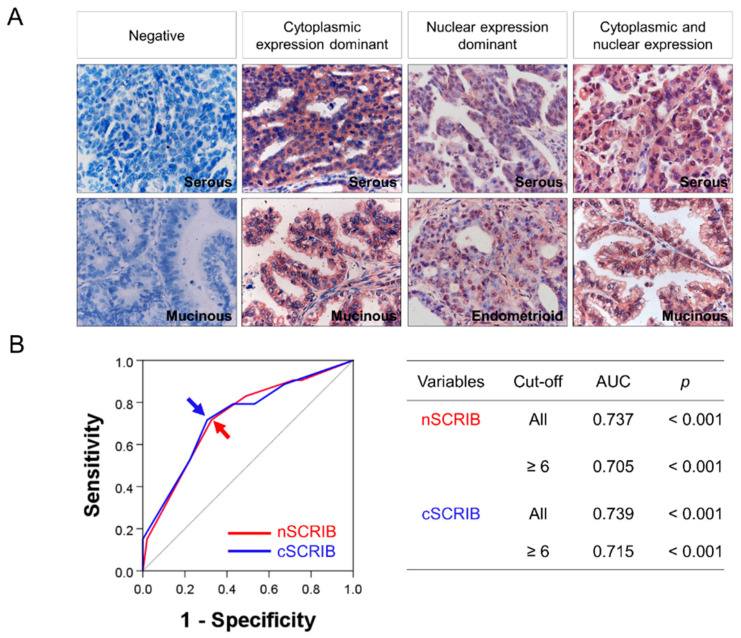
Immunohistochemical expression of SCRIB and statistical analysis. (**A**) SCRIB is expressed in both the cytoplasm and nuclei of ovarian carcinoma cells. Original magnification: ×400. (**B**) Receiver operating characteristic curve analysis to determine cut-off points for the nuclear expression of SCRIB (nSCRIB) and cytoplasmic expression of SCRIB (cSCRIB).

**Figure 2 biomolecules-11-00405-f002:**
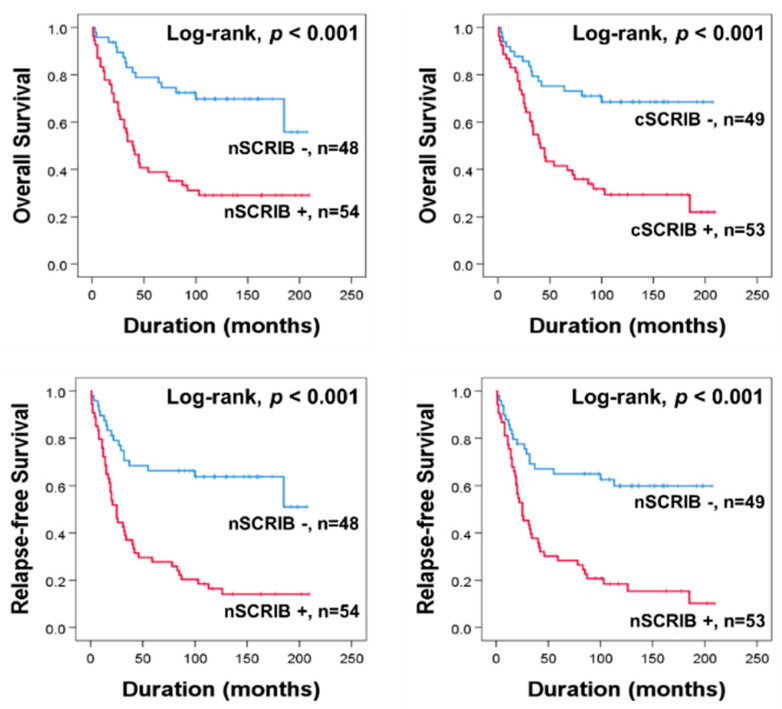
Kaplan-Meier survival curves according to the nuclear and cytoplasmic expression of SCRIB in 102 ovarian carcinoma patients.

**Figure 3 biomolecules-11-00405-f003:**
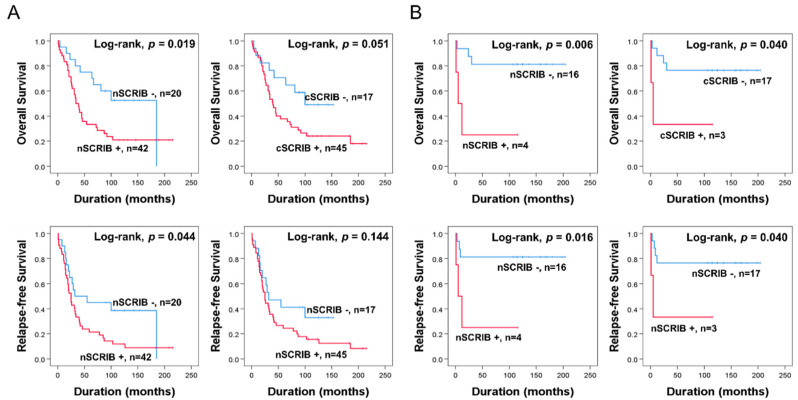
Kaplan-Meier survival curves in 62 HGSCs (**A**) and 20 mucinous carcinomas (**B**).

**Figure 4 biomolecules-11-00405-f004:**
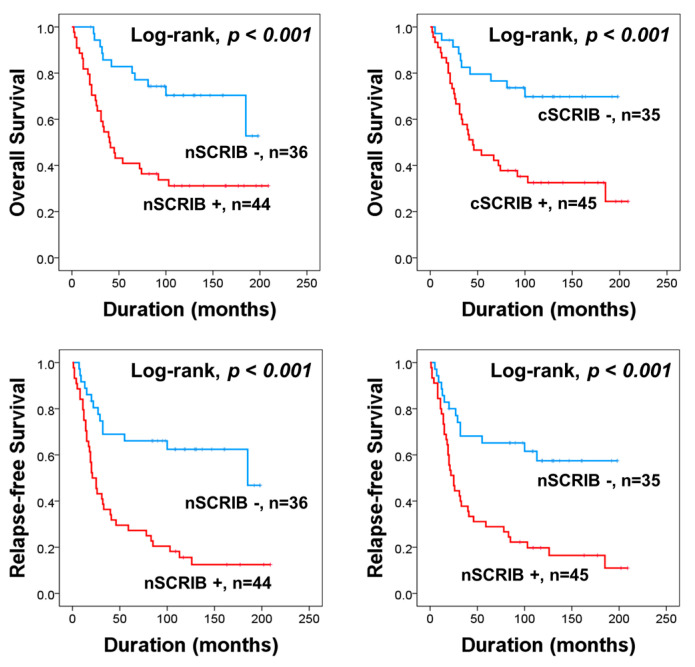
Survival analysis of ovarian carcinoma patients who received adjuvant chemotherapy. Kaplan-Meier survival curves for overall survival and relapse-free survival according to the nuclear and cytoplasmic expression of SCRIB in 80 ovarian carcinoma patients who received adjuvant chemotherapy.

**Figure 5 biomolecules-11-00405-f005:**
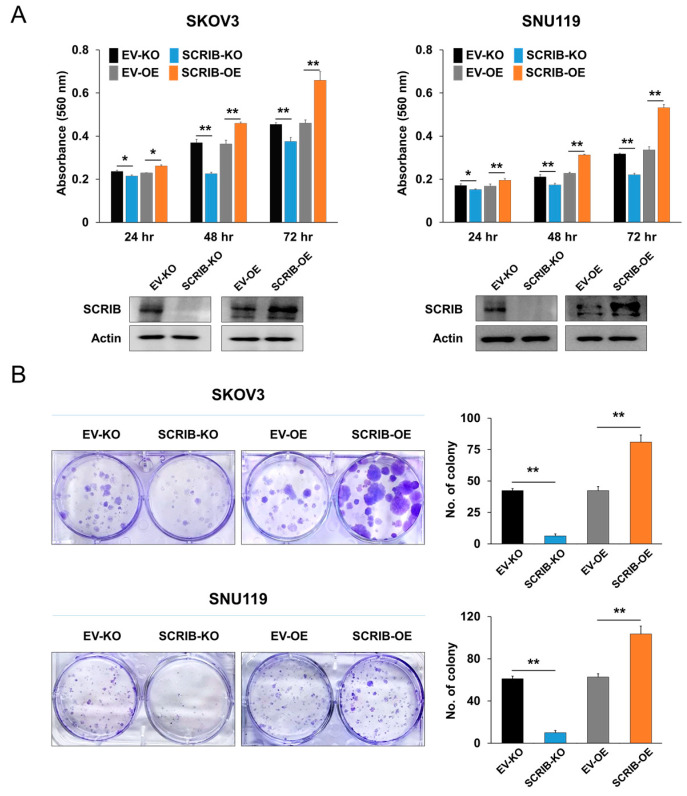
The effect of SCRIB on the proliferation of ovarian carcinoma cells. (**A**) SKOV3 and SNU119 cells were transfected with empty vector, vector for knockout of *SCRIB*, or SCRIB overexpression vector and performed CCK8 proliferation assay by growing 3000 cells in a 96-well plate for 24, 48, and 72 h. Knockout or overexpression of *SCRIB* in both SKOV3 and SNU119 cells were verified via western blot bands for SCRIB and actin. (**B**) The colony-forming assay in SKOV3 and SNU119 cells was performed by seeding 1000 cells per well in a 6-well plate for two weeks. * *p* < 0.05, ** *p* < 0.001.

**Figure 6 biomolecules-11-00405-f006:**
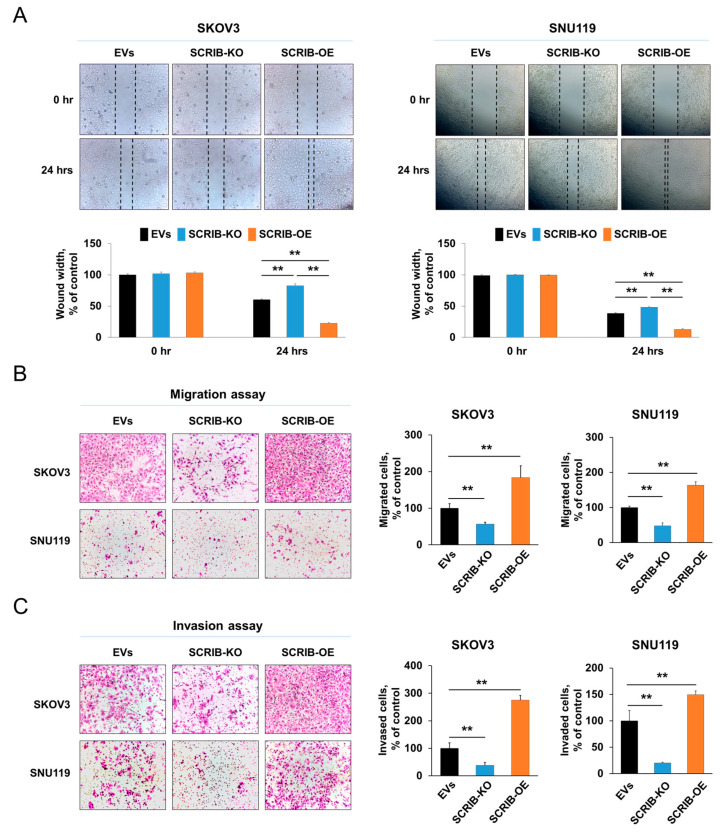
The effect of SCRIB on the invasiveness of ovarian carcinoma cells. (**A**) A wound healing assay was performed three times with SKOV3 and SNU119 cells transfected with empty vector, vector for knockout of *SCRIB*, or a *SCRIB* overexpression vector. EVs; the cells transfected with both empty vector for knockout and empty vector for overexpression of SCRIB. Microscopic images for wound healing assay were taken just after making linear scratches and again at 24 h after making scratches. The wound width was measured from the microscopic images and presented as ‘% of control’. The values in graphs are presented as the mean ± standard deviation. (**B**) The trans-chamber migration assay was performed by growing 5 × 10^4^ SKOV3 or 1 × 10^5^ SNU119 cells in the upper chamber for 48 h. (**C**) The trans-chamber invasion assay was performed by growing 1 × 10^5^ SKOV3 or 2 × 10^5^ SNU119 cells in the upper chamber with Matrigel for 48 h. The number of migrated or invaded cells were counted in five ×100 microscopic fields in each well. ** *p* < 0.001.

**Figure 7 biomolecules-11-00405-f007:**
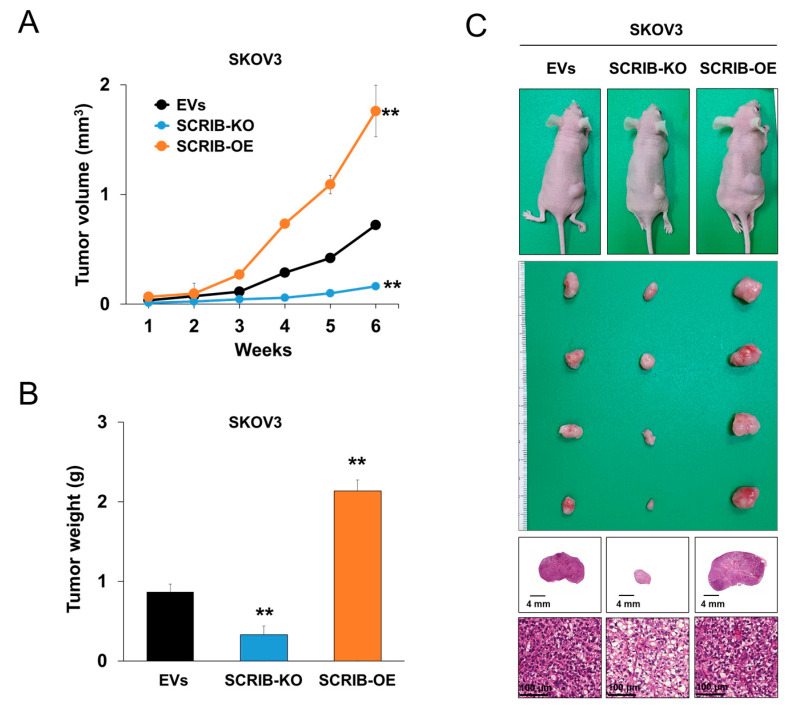
In vivo tumorigenic assay with knockout or overexpression of *SCRIB* in SKOV3 ovarian cancer cells. In vivo tumor growth was evaluated by subcutaneously implanting 2 × 10^6^ SKOV3 cells transfected with empty vectors, vector for knockout of *SCRIB*, or a *SCRIB* overexpression vector. EVs; the cells transfected with both empty vector for knockout and empty vector for overexpression of *SCRIB*. (**A**) The tumor volume was measured every week after tumor implantation by the equation V = L × W × H × 0.52 mm^3^. (**B**) Six weeks after tumor inoculation, mice were euthanized, and tumor weight was measured. (**C**) Macroscopic and microscopic findings of xenografted tumors. ** *p* < 0.001.

**Figure 8 biomolecules-11-00405-f008:**
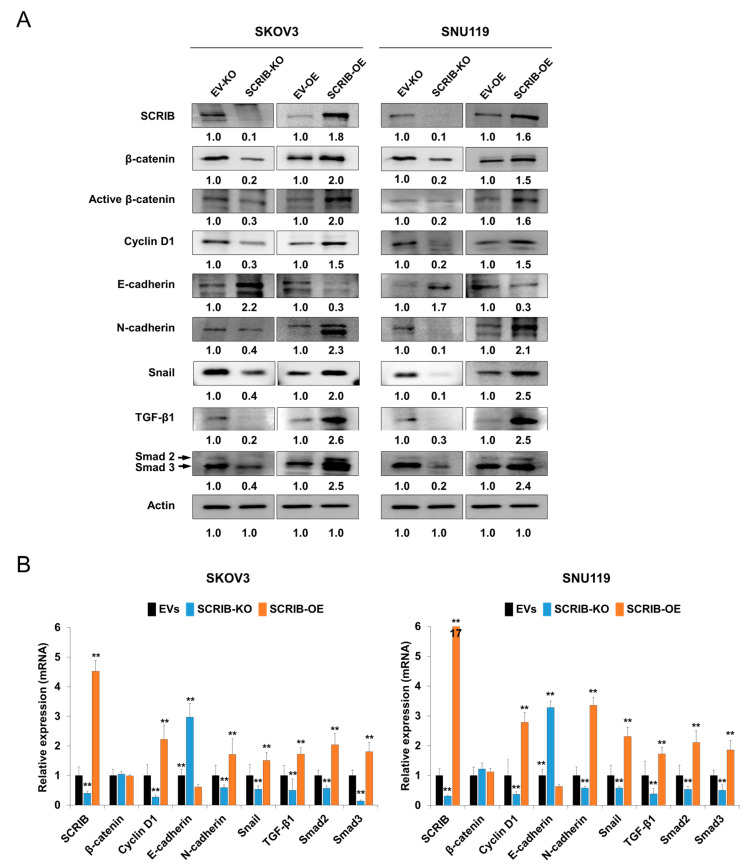
Western blot and quantitative reverse-transcription polymerase chain reaction after knockout or overexpression of *SCRIB* in ovarian carcinoma cells. (**A**) Western blot was performed for SCRIB, β-catenin, active β-catenin, cyclin D1, E-cadherin, N-cadherin, snail, TGF-β1, smad2/3, and actin after knockout or overexpression of *SCRIB* in SKOV3 and SNU119 ovarian carcinoma cells. The western bands were quantified using ImageJ software and the values are indicated below the bands. (**B**) Quantitative reverse-transcription polymerase chain reaction was performed for SCRIB, β-catenin, cyclin D1, E-cadherin, N-cadherin, snail, TGF-β1, smad2, and smad3 after knockout or overexpression of *SCRIB* in SKOV3 and SNU119 ovarian carcinoma cells. EVs; the cells transfected with both empty vector for knockout and empty vector for overexpression of *SCRIB*. ** *p* < 0.001.

**Table 1 biomolecules-11-00405-t001:** Primer sequences used for a qRT-PCR.

Gene	Primer Sequence		Product Size
*SCRIB*	forward	GGGACGACGAGGGCATATTC	207
	reverse	CGTTCTCAGGCTCCACCATGC	
*CTNNB1* (β-catenin)	forward	AAAATGGCAGTGCGTTTAG	100
	reverse	TTTGAAGGCAGTCTGTCGTA	
*CCND1* (Cyclin D1)	forward	GAGGAAGAGGAGGAGGAGGA	236
	reverse	GAGATGGAAGGGGGAAAGAG	
*E-cadherin*	forward	CCCGGGACAACGTTTATTAC	72
	reverse	ACTTCCCCTTCCTCAGTGAT	
*N-cadherin*	forward	ACAGTGGCCACCTACAAAGG	201
	reverse	CCGAGATGGGGTTGATAATG	
*TGF-* *β1*	forward	CCCACAACGAAATCTATGACAA	246
	reverse	AAGATAACCACTCTGGCGAGTG	
*SNAL1* (Snail)	forward	ACCCCACATCCTTCTCACTG	217
	reverse	TACAAAAACCCACGCAGACA	
*SMAD2* (Smad2)	forward	ACTAACTTCCCAGCAGGAAT	97
	reverse	GTTGGTCACTTGTTTCTCCA	
*SMAD3* (Smad3)	forward	CGAGAAATGGTGCGAGAAGG	259
	reverse	GAAGGCGAACTCACACAGC	
*GAPDH*	forward	AACAGCGACACCCACTCCTC	258
	reverse	GGAGGGGAGATTCAGTGTGGT	

**Table 2 biomolecules-11-00405-t002:** Clinicopathologic factors and the expression of SCRIB in ovarian carcinomas.

Characteristics		No.	nSCRIB		cSCRIB	
			Positive	*p*	Positive	*p*
Age, y	<60	69	32 (46%)	0.055	30 (43%)	0.013
	≥60	33	22 (67%)		23 (70%)	
CA125 *	Normal	18	3 (17%)	<0.001	3 (17%)	0.001
	Elevated	74	46 (62%)		44 (59%)	
Stage	I & II	52	21 (40%)	0.010	19 (37%)	0.001
	III & IV	50	33 (66%)		34 (68%)	
Cancer size, cm	≤10	67	40 (60%)	0.058	41 (61%)	0.010
	>10	35	14 (40%)		12 (34%)	
Lymph node metastasis	Absence	83	43 (52%)	0.632	42 (51%)	0.566
	Presence	19	11 (58%)		11 (58%)	
Ascites	Absence	69	30 (43%)	0.006	31 (45%)	0.040
	Presence	33	24 (73%)		22 (67%)	
Bilaterality	Unilateral	58	25 (43%)	0.022	25 (43%)	0.040
	Bilateral	44	29 (66%)		28 (64%)	
Histologic grade	Low (1)	26	4 (15%)	<0.001	4 (15%)	<0.001
	High (2 and 3)	76	50 (66%)		49 (64%)	
Histologic type	Serous	73	45 (62%)	0.015	49 (67%)	<0.001
	Mucinous	20	4 (20%)		3 (15%)	
	Endometrioid	5	3 (60%)		0 (0%)	
	Clear cell	3	2 (67%)		1 (33%)	
	Malignant Brenner	1	0 (0%)		0 (0%)	
Platinum-resistance	Absence	62	29 (47%)	0.006	31 (50%)	0.036
	Presence	18	15 (83%)		14 (78%)	
cSCRIB	Negative	49	6 (12%)	<0.001		
	Positive	53	48 (91%)			

* Pre-operative serum level of CA125 was not measured in 10 patients.

**Table 3 biomolecules-11-00405-t003:** Univariate analysis in overall ovarian carcinoma and 62 high-grade ovarian carcinomas.

Characteristics	No.	OS		RFS	
		HR (95% CI)	*p*	HR (95% CI)	*p*
Overall ovarian carcinomas (*n* = 104)					
Age, y, ≥60 (vs. <60)	33/102	2.763 (1.641–4.717)	<0.001	2.338 (1.438–3.802)	<0.001
Stage, III & IV (vs. I & II)	50/102	3.286 (1.845–5.853)	<0.001	4.206 (2.346–6.908)	<0.001
Cancer size, cm, >10 (vs. ≤10)	35/102	0.487 (0.262–0.906)	0.023	0.574 (0.334–0.985)	0.044
LN metastasis, presence (vs. absence)	19/102	1.320 (0.707–2.466)	0.383	1.697 (0.973–2.958)	0.062
Ascites, presence (vs. absence)	33/102	1.967 (1.154–3.353)	0.013	1.914 (1.170–3.131)	0.010
Bilaterality, bilateral (vs. unilateral)	44/102	1.642 (0.969–2.783)	0.065	2.115 (1.297–3.448)	0.003
CA125, elevated (vs. normal) *	74/92	5.083 (1.578–16.376)	0.006	5.009 (1.810–13.861)	0.002
Histologic grade, high (vs. low)	76/102	3.207 (1.444–7.124)	0.004	3.312 (1.631–6.725)	<0.001
nSCRIB, positive (vs. negative)	54/102	3.312 (1.820–6.026)	<0.001	3.606 (2.082–6.245)	<0.001
cSCRIB, positive (vs. negative)	53/102	3.179 (1.747–5.786)	<0.001	3.292 (1.917–5.655)	<0.001
High-grade serous carcinomas (*n* = 62)					
Age, y, ≥60 (vs. <60)	27/62	2.139 (1.170–3.911)	0.013	1.689 (0.972–2.934)	0.063
Stage, III & IV (vs. I & II)	37/62	1.822 (0.952–3.486)	0.070	2.831 (1.493–5.366)	0.001
Cancer size, cm, >10 (vs. ≤10)	14/62	0.703 (0.326–1.517)	0.370	0.773 (0.395–1.513)	0.452
LN metastasis, presence (vs. absence)	14/62	1.313 (0.658–2.621)	0.440	1.789 (0.953–3.356)	0.070
Ascites, presence (vs. absence)	26/62	1.342 (0.735–2.450)	0.338	1.391 (0.798–2.427)	0.245
Bilaterality, bilateral (vs. unilateral)	35/62	1.268 (0.684–2.350)	0.450	1.587 (0.887–2.840)	0.120
CA125, elevated (vs. normal)	52/57	2.373 (0.570–9.880)	0.235	1.589 (0.492–5.136)	0.439
nSCRIB, positive (vs. negative)	42/62	2.278 (1.117–4.644)	0.023	1.888 (1.002–3.560)	0.049
cSCRIB, positive (vs. negative)	45/62	2.116 (0.976–4.585)	0.058	1.635 (0.836–3.199)	0.151

* Pre-operative serum level of CA125 was not measured in 10 patients.

**Table 4 biomolecules-11-00405-t004:** Multivariate Cox regression analysis in overall ovarian carcinoma and 62 high-grade ovarian carcinomas.

Characteristics	OS		RFS	
	HR (95% CI)	*p*	HR (95% CI)	*p*
Overall ovarian carcinomas (*n* = 102) *				
Age, y, ≥60 (vs. <60)	2.254 (1.306–3.890)	0.004	1.654 (1.002–2.730)	0.049
Stage, III & IV (vs. I & II)	2.218 (1.212–4.059)	0.010	2.802 (1.573–4.990)	<0.001
Histologic grade, high (vs. low)			2.353 (1.071–5.168)	0.033
nSCRIB, positive (vs. negative)	2.252 (1.205–4.207)	0.011	1.768 (0.963–3.245)	0.066
High-grade serous carcinomas (*n* = 62) **				
Age, y, ≥60 (vs. <60)	2.103 (1.150–3.843)	0.016		
Stage, III & IV (vs. I & II)			2.831 (1.493–5.366)	0.001
nSCRIB, positive (vs. negative)	2.236 (1.097–4.558)	0.027		

* The factors included in the multivariate analysis were age, cancer stage, cancer size, ascites, bilaterality, histologic grade, nSCRIB, and cSCRIB. ** The factors included in the multivariate analysis were age, cancer stage, cancer size, ascites, bilaterality, histologic grade, nSCRIB, and cSCRIB.

**Table 5 biomolecules-11-00405-t005:** Univariate and multivariate analysis in 80 ovarian carcinoma patients who received adjuvant chemotherapy.

Characteristics	No.	OS		RFS	
		HR (95% CI)	*p*	HR (95% CI)	*p*
Univariate analysis					
Age, y, ≥60 (vs. <60)	26/80	2.805 (1.527–5.153)	0.001	2.152 (1.248–3.713)	0.006
Stage, III & IV (vs. I & II)	44/80	3.213 (1.605–6.433)	<0.001	3.709 (1.984–6.934)	<0.001
Cancer size, cm, >10 (vs. ≤10)	26/80	0.617 (0.303–1.255)	0.182	0.766 (0.421–1.394)	0.383
LN metastasis, presence (vs. absence)	17/80	1.169 (0.572–2.391)	0.668	1.489 (0.805–2.756)	0.205
Ascites, presence (vs. absence)	29/80	2.306 (1.252–4.247)	0.007	2.045 (1.182–3.537)	0.010
Bilaterality, bilateral (vs. unilateral)	37/80	1.465 (0.796–2.697)	0.220	1.933 (1.113–3.358)	0.019
CA125, elevated (vs. normal)	65/77	10.560 (1.448–77.004)	0.020	7.140 (1.731–29.448)	0.007
Histologic grade, high (vs. low)	62/80	5.150 (1.140–23.276)	0.033	2.693 (1.210–5.991)	0.015
nSCRIB, positive (vs. negative)	44/80	3.352 (1.680–6.688)	<0.001	3.598 (1.943–6.664)	<0.001
cSCRIB, positive (vs. negative)	45/80	3.334 (1.634–6.801)	<0.001	3.182 (1.719–5.890)	<0.001
Multivariate analysis					
Age, y, ≥60 (vs. <60)		2.308 (1.244–4.280)	0.008		
Stage, III & IV (vs. I & II)		4.079 (1.672–9.953)	0.002	2.619 (1.343–5.108)	0.005
Bilaterality, bilateral (vs. unilateral)		0.436 (0.202–0.942)	0.035		
nSCRIB, positive (vs. negative)		2.363 (1.149–4.859)	0.019	2.530 (1.313–4.875)	0.006

Age, cancer stage, cancer size, ascites, bilaterality, histologic grade, nSCRIB, and cSCRIB were included in multivariate analysis.

## Data Availability

The datasets used in the current study are available from the corresponding author upon reasonable request.

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
