# Peer review of "SCRIB Is Involved in the Progression of Ovarian Carcinomas in Association with the Factors Linked to Epithelial-to-Mesenchymal Transition and Predicts Shorter Survival of Diagnosed Patients"

_biomolecules, 2021, doi:10.3390/biom11030405_

Round 1

Reviewer 1 Report

Summary: Hussein et al. described the expression of SCRIB in ovarian cancer patient samples, its association with patient survival and other patient characteristics. They also perform gain- and loss-of-function studies in two cell lines to examine the effect on proliferation and invasion. Finally, they examine the results of gain and loss of SCRIB on expression of factors associated with EMT on protein and mRNA levels.

Major points:

  • Please provide more explanation for the results in Fig. 1. Line 175: it is unclear what the meaning of the following sentence is: “The cut-off points were six in both nSCRIB and cSCRIB”.

  • Figure 4 is missing, although it is referred to in Line 218.

  • Are the results in Fig. 5 representative of a single experiment? How many technical / biological replicates? Are error bars standard deviation? Same questions for results in Fig. 6,7,8.

  • Please describe bottom panels of Fig. 5A.

  • On Fig. 6A, please quantify the change in wound healing with KO or OE of SCRIB.

  • Please provide quantification of WB results.

  • Please provide images of whole blots for all WB data.

  • Methods descriptions are incomplete: e.g.
    1. List all media components
    2. Include the concentration of Matrigel used
    3. Include identifying information (clone or catalog #) for antibodies used
    4. Indicate the analysis method used for qPCR results (delta delta cT?)
    5. Did mice reach endpoints in the in vivo tumorigenic assay? Why was the six week time point chosen?
    6. For the colony forming assay, how were cells fixed?

  • Claims that SCRIB is associated with EMT should be supported with morphological evidence such as a change from cobblestone to spindle shape, or a change in circularity, etc. Otherwise it would be more accurate to state that SCRIB is associated with a change in expression of factors linked to EMT.

Minor points: 

  • Line 104: “knock-down” should be knockout (also line 230, etc. and throughout the manuscript)
  • In Fig. 5 and 8, two separate control vectors are used, one for knockout, one for overexpression. However, in Fig. 6 and 7, only one condition for ‘EVs’ is used. Please explain.
  • Please follow protein naming conventions: e.g. line 252 when referring to gene, gene name should be italicized.
  • The authors’ use of the term ‘cytoplasmic membrane’ is confusing. A more conventional term would add clarity; e.g. plasma membrane (as used in ref. 7) or plasmalemma.
  • Delete the word ‘was’ on line 317.
  • Table 3: what does the asterisk indicate?
  • Please give a reference for the formula used for calculation of tumor size.
  • I presume SCRIN on line 252 should be SCRIB?
  • Is it correct that the tumor weight shown in Fig. 7B is mg? Not grams?

Reviewer 2 Report

Hussein et al. report a potential contribution of SCRIB to ovarian cancer malignancy. Specifically, they show how nuclear staining of SCRIB in biopsies could be a predictor for prognosis in patients. The study is well designed and the conclusions regarding SCRIB as a potential predictor have merit, although they will need to be further validated with much bigger cohorts.

However, this reviewer considers that there are some points that need to be addressed.

Major point:

  • The experiments where authors overexpress or knock down SCRIB show that it could have an impact on EMT. This would mean that it most likely influence metastasis. However, the in vivo experiment chosen for its validation is designed only to address tumor growth. It would be much more interesting if the author could perform an experiment where the role of SCRIB in metastasis is evaluated. This could be achieved by orthotopic injection of tumor cells in mouse ovaries and evaluating the spread into the peritoneal cavity. Additionally, Resection of primary tumors to explore the metastatic potential in the absence of a primary tumor could also be of interest.

Minor points:

  • There is no Figure 4, this should be corrected.
  • In Figure 5 the images of the trans-chamber invasion assay do not represent the quantification on the bar graph. For example, 5B. It is much bigger the effect observed in the picture as compared to that on the graph. Authors should choose pictures that better represent the mean shown in the graph.
  • Some of the results depicted in Figure 8 are difficult to understand. The result regarding SMADs is confusing because SMADs are degraded upon activation, so it could also mean that this signaling pathway is inhibited in reality. It would be advisable to evaluate phosphorylation of these proteins. It would also be informative if authors could evaluate SMAD7 expression as the main inhibitor of this signaling pathway. In addition, TGF-β western blot is of a very bad quality and it would need to be confirmed by ELISA.

Round 2

Reviewer 1 Report

Summary: The authors have responded to all points raised in review. A few issues remain, as follows. If these points can be adequately addressed, the manuscript may be suitable for publication in Biomolecules.

Major points:

  • Comment on previous point 3: The authors’ response that the experiments were done in triplicate does not address the question of how many biological vs. technical replicates were done. The assumption from their response is that three technical replicates (one biological replicate) were done. In general experiments should repeated (biological replicates) at least three times, and therefore without further clarification it does not appear that these results are robust enough for publication in Biomolecules.
  • Comment on previous point 4: Why is the amount of SCRIB as shown in the WB for Fig. 5A so different in the two WB? In the EV-KO, there is a large amount of SCRIB, which decreases as expected in the KO; in the EV-OE the SCRIB expression is negligible, and increases to a similar level as EV-KO with OE. Amount of SCRIB in EV-KO and EV-OE should be similar.
  • Comment on previous point 5: Please describe how the quantitation of the migration assay was done. How many times was the experiment done? How many measurements were taken, on how many scratches? (In other words, what does the standard deviation as shown by error bars represent?)
  • Comment on previous point 8e: Please add a sentence to Materials and Methods describing that mice were euthanized according to humane end points at 6 weeks.
  • Comment on previous point 9: Although the authors changed the heading of 3.6 to indicate ‘SCRIB is associated with a change in expression of factors linked to the EMT of cancer cells’ instead of ‘SCRIB is associated with EMT of cancer cells’, the title of the manuscript still indicates that SCRIB is associated with EMT. The title should be modified to avoid this claim. Related to this point, in the discussion (line 419), stating that SCRIB stimulated EMT is not substantiated by the data; it can only be stated that the induction of EMT-related factors correlated with SCRIB expression. Similarly, in line 435-437, SCRIB can be said to induce some EMT characteristics. Again in line 478, 481, please modify these statements.

Author Response

Response to reviewer 1

We thank the reviewer for these insightful comments.

Reviewer reports:  Comments and Suggestions for Authors

Summary: The authors have responded to all points raised in review. A few issues remain, as follows. If these points can be adequately addressed, the manuscript may be suitable for publication in Biomolecules.

 We thank the reviewer for this comment.

Major points:

Comment on previous point 3: The authors’ response that the experiments were done in triplicate does not address the question of how many biological vs. technical replicates were done. The assumption from their response is that three technical replicates (one biological replicate) were done. In general experiments should repeated (biological replicates) at least three times, and therefore without further clarification it does not appear that these results are robust enough for publication in Biomolecules.

 We thank the reviewer for this comment. All experiments were performed three times, and representative data are presented with the mean ± standard deviation. In response to the comment of the reviewer, we have revised the manuscript to make this clear. Below are the revised sentences in the manuscript.

All experiments were done in triplicate and performed three times, with representative data presented.

Comment on previous point 4: Why is the amount of SCRIB as shown in the WB for Fig. 5A so different in the two WB? In the EV-KO, there is a large amount of SCRIB, which decreases as expected in the KO; in the EV-OE the SCRIB expression is negligible, and increases to a similar level as EV-KO with OE. Amount of SCRIB in EV-KO and EV-OE should be similar.

 We thank the reviewer for this comment. In our experiment, the images of western blot bands were detected using LAS-3000 digital imaging detection system. Therefore, the thickness of western band might be different according to the duration of exposure time, and we focused to show a relatively higher expression of SCRIB when transfected with SCRIB-overexpression vector than when transfected with empty vector. However, we agree with the reviewer. Therefore, in response to the comment of the reviewer, we have revised Figures 5A and 8A to present clear western blot bands of SCRIB.

Comment on previous point 5: Please describe how the quantitation of the migration assay was done. How many times was the experiment done? How many measurements were taken, on how many scratches? (In other words, what does the standard deviation as shown by error bars represent?)

 We thank the reviewer for this comment. The wound healing assay was performed three times, and we took images. Thereafter, we measured the width of the wound and the data presented as ‘% of control’. Therefore, the error bar in Figure 6A is derived from the standard deviation of data obtained from the three times experiments of wound healing assay. In response to the comment of the reviewer, we have revised the manuscript to make this clear. Below are the revised sentences in the manuscript.

Figure 6. The effect of SCRIB on the invasiveness of ovarian carcinoma cells. (A) A wound healing assay was performed three times with SKOV3 and SNU119 cells transfected with empty vector, vector for knockout of SCRIB, or a SCRIB overexpression vector. EVs; the cells transfected with both empty vector for knockout and empty vector for overexpression of SCRIB. Microscopic images for wound healing assay were taken just after making linear scratches and again at 24 hours after making scratches. The wound width was measured from the microscopic images and presented as ‘% of control’. The values in graphs are presented as the mean ± standard deviation.

Comment on previous point 8e: Please add a sentence to Materials and Methods describing that mice were euthanized according to humane end points at 6 weeks.

 We thank the reviewer for this comment. In response to the comment of the reviewer, we have revised the manuscript. Below are the revised sentences in the manuscript.

The mice were euthanized according to humane end points at six weeks after tumor cell inoculation.

Comment on previous point 9: Although the authors changed the heading of 3.6 to indicate ‘SCRIB is associated with a change in expression of factors linked to the EMT of cancer cells’ instead of ‘SCRIB is associated with EMT of cancer cells’, the title of the manuscript still indicates that SCRIB is associated with EMT. The title should be modified to avoid this claim. Related to this point, in the discussion (line 419), stating that SCRIB stimulated EMT is not substantiated by the data; it can only be stated that the induction of EMT-related factors correlated with SCRIB expression. Similarly, in line 435-437, SCRIB can be said to induce some EMT characteristics. Again in line 478, 481, please modify these statements.

 We thank the reviewer for this comment. In response to the comment of the reviewer, we have revised the manuscript. Below are the revised sentences in the manuscript.

Title

SCRIB is involved in the progression of ovarian carcinomas in association with the factors linked to epithelial-to-mesenchymal transition and predicts shorter survival of diagnosed patients

Discussion and conclusion section

In our results, SCRIB stimulated invasiveness and the factors linked to EMT of ovarian cancer cells.

In addition, SCRIB induced expression of factors linked to EMT phenotype and increased invasion of gastric cancer cells [10] and cytoplasmic SCRIB increased the invasion activity of liver cancer cells [7]. Therefore, these results suggest that SCRIB is involved in the progression of ovarian cancer by inducing the expression of factors linked to EMT.

In this study, we demonstrated that SCRIB stimulates proliferation and invasiveness of ovarian cancer cells in conjunction with activation of the factors linked to the EMT pathway.

Therefore, our results suggest that SCRIB is involved in cancer progression and chemoresistance by activating some EMT characteristics.

Reviewer 2 Report

None

Author Response

We thank the reviewer.